# Unbiased estimates for linear regression via volume sampling

**Michał Dereziński**
Department of Computer Science
University of California Santa Cruz
mderezin@ucsc.edu

**Manfred K. Warmuth**
Department of Computer Science
University of California Santa Cruz
manfred@ucsc.edu

## Abstract

Given a full rank matrix $\mathbf{X}$ with more columns than rows, consider the task of estimating the pseudo inverse $\mathbf{X}^+$ based on the pseudo inverse of a sampled subset of columns (of size at least the number of rows). We show that this is possible if the subset of columns is chosen proportional to the squared volume spanned by the rows of the chosen submatrix (ie, volume sampling). The resulting estimator is unbiased and surprisingly the covariance of the estimator also has a closed form: It equals a specific factor times $\mathbf{X}^{+\top}\mathbf{X}^+$.

Pseudo inverse plays an important part in solving the linear least squares problem, where we try to predict a label for each column of $\mathbf{X}$. We assume labels are expensive and we are only given the labels for the small subset of columns we sample from $\mathbf{X}$. Using our methods we show that the weight vector of the solution for the sub problem is an unbiased estimator of the optimal solution for the whole problem based on all column labels.

We believe that these new formulas establish a fundamental connection between linear least squares and volume sampling. We use our methods to obtain an algorithm for volume sampling that is faster than state-of-the-art and for obtaining bounds for the total loss of the estimated least-squares solution on all labeled columns.

## 1 Introduction

Let $\mathbf{X}$ be a wide full rank matrix with $d$ rows and $n$ columns where $n \geq d$. Our goal is to estimate the pseudo inverse $\mathbf{X}^+$ of $\mathbf{X}$ based on the pseudo inverse of a subset of columns. More precisely, we sample a subset $S \subseteq \{1..n\}$ of $s$ column indices (where $s \geq d$). We let $\mathbf{X}_S$ be the sub-matrix of the $s$ columns indexed by $S$ (See Figure 1). Consider a version of $\mathbf{X}$ in which all but the columns of $S$ are zero. This matrix equals $\mathbf{X}\mathbf{I}_S$ where $\mathbf{I}_S$ is an $n$-dimensional diagonal matrix with $(\mathbf{I}_S)_{ii} = 1$ if $i \in S$ and 0 otherwise.

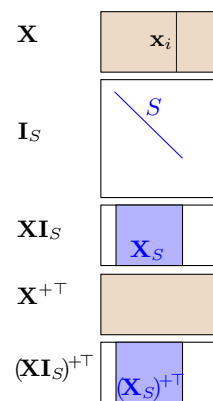

**Figure 1:** Set $S$ may not be consecutive.

We assume that the set of $s$ column indices of $\mathbf{X}$ is selected proportional to the squared volume spanned by the rows of submatrix $\mathbf{X}_S$, i.e. proportional to $\det(\mathbf{X}_S\mathbf{X}_S^\top)$ and prove a number of new surprising expectation formulas for this type of volume sampling, such as

$$\mathbb{E}[(\mathbf{X}\mathbf{I}_S)^+] = \mathbf{X}^+ \quad \text{and} \quad \mathbb{E}[\ \underbrace{(\mathbf{X}_S\mathbf{X}_S^\top)^{-1}}_{(\mathbf{X}\mathbf{I}_S)^{+\top}(\mathbf{X}\mathbf{I}_S)^+}\ ] = \frac{n-d+1}{s-d+1}\ \mathbf{X}^{+\top}\mathbf{X}^+.$$

Note that $(\mathbf{X}\mathbf{I}_S)^+$ has the $n \times d$ shape of $\mathbf{X}^+$ where the $s$ rows indexed by $S$ contain $(\mathbf{X}_S)^+$ and the remaining $n-s$ rows are zero. The expectation of this matrix is $\mathbf{X}^+$ even though $(\mathbf{X}_S)^+$ is

clearly not a sub-matrix of $\mathbf{X}^+$. In addition to the expectation formulas, our new techniques lead to an efficient volume sampling procedure which beats the state-of-the-art by a factor of $n^2$ in time complexity.

Volume sampling is useful in numerous applications, from clustering to matrix approximation, but we focus on the task of solving linear least squares problems: For an $n-$dimensional label vector $\mathbf{y}$, let $\mathbf{w}^* = \operatorname{argmin}_{\mathbf{w}} ||\mathbf{X}^\top \mathbf{w} - \mathbf{y}||^2 = \mathbf{X}^+ \mathbf{y}$. Assume the entire design matrix $\mathbf{X}$ is known to the learner but labels are expensive and you want to observe as few of them as possible. Let $\mathbf{w}^*(S) = (\mathbf{X}_S)^+ \mathbf{y}_S$ be the solution to the sub-problem based on labels $\mathbf{y}_S$. What is the smallest number of labels $s$ necessary, for which there is a sampling procedure on sets $S$ of size $s$ st the expected loss of $\mathbf{w}^*(S)$ is at most a constant factor larger than the loss of $\mathbf{w}^*$ that uses all $n$ labels (where the constant is independent of $n$)? More precisely, using the short hand $L(\mathbf{w}) = ||\mathbf{X}^\top \mathbf{w} - \mathbf{y}||^2$ for the loss on all $n$ labels, what is the smallest size $s$ such that $\mathbb{E}[L(\mathbf{w}^*(S))] \leq \operatorname{const} L(\mathbf{w}^*)$. This question is a version of the "minimal coresets" open problem posed in [3].

The size has to be at least $d$ and one can show that randomization is necessary in that any deterministic algorithm for choosing a set of $d$ columns can suffer loss larger by a factor of $n$. Also any iid sampling of $S$ (such as the commonly used leverage scores [8]) requires at least $\Omega(d \log d)$ examples to achieve a finite factor. In this paper however we show that with a size $d$ volume sample, $\mathbb{E}[L(\mathbf{w}^*(S))] = (d+1)L(\mathbf{w}^*)$ if $\mathbf{X}$ is in general position. Note again that we have equality and not just an upper bound. Also we can show that the multiplicative factor $d+1$ is optimal. We further improve this factor to $1 + \epsilon$ via repeated volume sampling. Moreover, our expectation formulas imply that when $S$ is size $s \geq d$ volume sampled, then $\mathbf{w}^*(S)$ is an unbiased estimator for $\mathbf{w}^*$, ie $\mathbb{E}[\mathbf{w}^*(S)] = \mathbf{w}^*$.

## 2  Related work

Volume sampling is an extension of a determinantal point process [15], which has been given a lot of attention in the literature with many applications to machine learning, including recommendation systems [10] and clustering [13]. Many exact and approximate methods for efficiently generating samples from this distribution have been proposed [6, 14], making it a useful tool in the design of randomized algorithms. Most of those methods focus on sampling $s \leq d$ elements. In this paper, we study volume sampling sets of size $s \geq d$, which has been proposed in [1] and motivated with applications in graph theory, linear regression, matrix approximation and more. The only known polynomial time algorithm for size $s > d$ volume sampling was recently proposed in [16] with time complexity $O(n^4 s)$. We offer a new algorithm with runtime $O((n - s + d)nd)$, which is faster by a factor of at least $n^2$.

The problem of selecting a subset of input vectors for solving a linear regression task has been extensively studied in statistics literature under the terms *optimal design* [9] and *pool-based active learning* [19]. Various criteria for subset selection have been proposed, like A-optimality and D-optimality. For example, A-optimality seeks to minimize $\operatorname{tr}((\mathbf{X}_S \mathbf{X}_S^\top)^{-1})$, which is combinatorially hard to optimize exactly. We show that for size $s$ volume sampling (for $s \geq d$), $\mathbb{E}[(\mathbf{X}_S \mathbf{X}_S^\top)^{-1}] = \frac{n-d+1}{s-d+1} \mathbf{X}^{+\top} \mathbf{X}^+$ which provides an approximate randomized solution for this task.

A related task has been explored in the field of computational geometry, where efficient algorithms are sought for approximately solving linear regression and matrix approximation [17, 5, 3]. Here, multiplicative bounds on the loss of the approximate solution can be achieved via two approaches: Subsampling the vectors of the design matrix, and sketching the design matrix $\mathbf{X}$ and the label vector $\mathbf{y}$ by multiplying both by the same suitably chosen random matrix. Algorithms which use sketching to generate a smaller design matrix for a given linear regression problem are computationally efficient [18, 5], but unlike vector subsampling, they require all of the labels from the original problem to generate the sketch, so they do not apply directly to our setting of using as few labels as possible.

The main competitor to volume sampling for linear regression is iid sampling using the statistical leverage scores [8]. However we show in this paper that any iid sampling method requires sample size $\Omega(d \log d)$ to achieve multiplicative loss bounds. On the other hand, the input vectors obtained from volume sampling are *selected jointly* and this makes the chosen subset more informative. We show that just $d$ volume sampled columns are sufficient to achieve a multiplicative bound. Volume sampling size $s \leq d$ has also been used in this line of work by [7, 11] for matrix approximation.

# 3 Unbiased estimators

Let $n$ be an integer dimension. For each subset $S \subseteq \{1..n\}$ of size $s$ we are given a matrix formula $\mathbf{F}(S)$. Our goal is to sample set $S$ of size $s$ using some sampling process and then develop concise expressions for $\mathbb{E}_{S:|S|=s}[\mathbf{F}(S)]$. Examples of formula classes $\mathbf{F}(S)$ will be given below.

We represent the sampling by a directed acyclic graph (dag), with a single root node corresponding to the full set $\{1..n\}$, Starting from the root, we proceed along the edges of the graph, iteratively removing elements from the set $S$. Concretely, consider a dag with levels $s = n, n-1, ..., d$. Level $s$ contains $\binom{n}{s}$ nodes for sets $S \subseteq \{1..n\}$ of size $s$. Every node $S$ at level $s > d$ has $s$ directed edges to the nodes $S - \{i\}$ at the next lower level. These edges are labeled with a conditional probability vector $P(S_{-i}|S)$. The probability of a (directed) path is the product of the probabilities along its edges. The outflow of probability from each node on all but the bottom level is 1. We let the probability $P(S)$ of node $S$ be the probability of all paths from the top node $\{1..n\}$ to $S$ and set the probability $P(\{1..n\})$ of the top node to 1. We associate a formula $\mathbf{F}(S)$ with each set node $S$ in the dag. The following key equality lets us compute expectations.

**Lemma 1** *If for all $S \subseteq \{1..n\}$ of size greater than $d$ we have*

$$\mathbf{F}(S) = \sum_{i \in S} P(S_{-i}|S)\mathbf{F}(S_{-i}),$$

*then for any $s \in \{d..n\}$:* $\mathbb{E}_{S:|S|=s}[\mathbf{F}(S)] = \sum_{S:|S|=s} P(S)\mathbf{F}(S) = \mathbf{F}(\{1..n\})$.

**Proof** Suffices to show that expectations at successive layers are equal:

$$\sum_{S:|S|=s} P(S)\,\mathbf{F}(S) = \sum_{S:|S|=s} P(S)\sum_{i \in S} P(S_{-i}|S)\,\mathbf{F}(S_{-i}) = \sum_{T:|T|=s-1} \underbrace{\sum_{j \notin T} P(T_{+j})P(T|T_{+j})}_{P(T)}\mathbf{F}(T). \quad \blacksquare$$

## 3.1 Volume sampling

Given a wide full-rank matrix $\mathbf{X} \in \mathbb{R}^{d \times n}$ and a sample size $s \in \{d..n\}$, volume sampling chooses subset $S \subseteq \{1..n\}$ of size $s$ with probability proportional to volume spanned by the rows of submatrix $\mathbf{X}_S$, ie proportional to $\det(\mathbf{X}_S\mathbf{X}_S^\top)$. The following corollary uses the above dag setup to compute the normalization constant for this distribution. When $s = d$, the corollary provides a novel minimalist proof for the Cauchy-Binet formula: $\sum_{S:|S|=s} \det(\mathbf{X}_S\mathbf{X}_S^\top) = \det(\mathbf{X}\mathbf{X}^\top)$.

**Corollary 2** *Let $\mathbf{X} \in \mathbb{R}^{d \times n}$ and $S \subseteq \{1..n\}$ of size $n \geq s \geq d$ st $\det(\mathbf{X}_S\mathbf{X}_S^\top) > 0$. Then for any set $S$ of size larger than $d$ and $i \in S$, define the probability of the edge from $S$ to $S_{-i}$ as:*

$$P(S_{-i}|S) := \frac{\det(\mathbf{X}_{S_{-i}}\mathbf{X}_{S_{-i}}^\top)}{(s-d)\det(\mathbf{X}_S\mathbf{X}_S^\top)} = \frac{1 - \mathbf{x}_i^\top (\mathbf{X}_S\mathbf{X}_S^\top)^{-1}\mathbf{x}_i}{s-d}, \quad \textbf{(reverse iterative volume sampling)}$$

*where $\mathbf{x}_i$ is the $i$th column of $\mathbf{X}$ and $\mathbf{X}_S$ is the sub matrix of columns indexed by $S$. Then $P(S_{-i}|S)$ is a proper probability distribution and thus $\sum_{S:|S|=s} P(S) = 1$ for all $s \in \{d..n\}$. Furthermore*

$$P(S) = \frac{\det(\mathbf{X}_S\mathbf{X}_S^\top)}{\binom{n-d}{s-d}\det(\mathbf{X}\mathbf{X}^\top)}. \quad \textbf{(volume sampling)}$$

**Proof** First, for any node $S$ st $s > d$ and $\det(\mathbf{X}_S\mathbf{X}_S^\top) > 0$, the probabilities out of $S$ sum to 1:

$$\sum_{i \in S} P(S_{-i}|S) = \sum_{i \in S} \frac{1 - \text{tr}((\mathbf{X}_S\mathbf{X}_S^\top)^{-1}\mathbf{x}_i\mathbf{x}_i^\top)}{s-d} = \frac{s - \text{tr}((\mathbf{X}_S\mathbf{X}_S^\top)^{-1}\mathbf{X}_S\mathbf{X}_S^\top)}{s-d} = \frac{s-d}{s-d} = 1.$$

It remains to show the formula for the probability $P(S)$ of all paths ending at node $S$. Consider any path from the root $\{1..n\}$ to $S$. There are $(n-s)!$ such paths. The fractions of determinants in

probabilities along each path telescope[1] and the additional factors accumulate to the same product. So the probability of all paths from the root to $S$ is the same and the total probability into $S$ is

$$\frac{(n-s)!}{(n-d)(n-d-1)\ldots(n-s+1)}\,\frac{\det(\mathbf{X}_S\mathbf{X}_S^\top)}{\det(\mathbf{X}\mathbf{X}^\top)} \;=\; \frac{1}{\binom{n-d}{s-d}}\,\frac{\det(\mathbf{X}_S\mathbf{X}_S^\top)}{\det(\mathbf{X}\mathbf{X}^\top)}. \qquad \blacksquare$$

### 3.2 Expectation formulas for volume sampling

All expectations in the remainder of the paper are wrt volume sampling. We use the short hand $\mathbb{E}[\mathbf{F}(S)]$ for expectation with volume sampling where the size of the sampled set is fixed to $s$. The expectation formulas for two choices of $\mathbf{F}(S)$ are proven in the next two theorems. By Lemma 1 it suffices to show $\mathbf{F}(S) = \sum_{i \in S} P(S_{-i}|S)\mathbf{F}(S_{-i})$ for volume sampling.

We introduce a bit more notation first. Recall that $\mathbf{X}_S$ is the sub matrix of columns indexed by $S \subseteq \{1..n\}$ (See Figure 1). Consider a version of $\mathbf{X}$ in which all but the columns of $S$ are zero. This matrix equals $\mathbf{X}\mathbf{I}_S$ where $\mathbf{I}_S$ is an $n$-dimensional diagonal matrix with $(\mathbf{I}_S)_{ii} = 1$ if $i \in S$ and 0 otherwise.

**Theorem 3** *Let $\mathbf{X} \in \mathbb{R}^{d \times n}$ be a wide full rank matrix (ie $n \geq d$). For $s \in \{d..n\}$, let $S \subseteq 1..n$ be a size $s$ volume sampled set over $\mathbf{X}$. Then*

$$\mathbb{E}[(\mathbf{X}\mathbf{I}_S)^+] = \mathbf{X}^+.$$

We believe that this fundamental formula lies at the core of why volume sampling is important in many applications. In this work, we focus on its application to linear regression. However, [1] discuss many problems where controlling the pseudo-inverse of a submatrix is essential. For those applications, it is important to establish variance bounds for the estimator offered by Theorem 3. In this case, volume sampling once again offers very concrete guarantees. We obtain them by showing the following formula, which can be viewed as a second moment for this estimator.

**Theorem 4** *Let $\mathbf{X} \in \mathbb{R}^{d \times n}$ be a full-rank matrix and $s \in \{d..n\}$. If size $s$ volume sampling over $\mathbf{X}$ has full support, then*

$$\mathbb{E}[\underbrace{(\mathbf{X}_S\mathbf{X}_S^\top)^{-1}}_{(\mathbf{X}\mathbf{I}_S)^{+\top}(\mathbf{X}\mathbf{I}_S)^+}] = \frac{n-d+1}{s-d+1}\underbrace{(\mathbf{X}\mathbf{X}^\top)^{-1}}_{\mathbf{X}^{+\top}\mathbf{X}^+}.$$

*If volume sampling does not have full support then the matrix equality "=" is replaced by the positive-definite inequality "$\preceq$".*

The condition that size $s$ volume sampling over $\mathbf{X}$ has full support is equivalent to $\det(\mathbf{X}_S\mathbf{X}_S^\top) > 0$ for all $S \subseteq 1..n$ of size $s$. Note that if size $s$ volume sampling has full support, then size $t > s$ also has full support. So full support for the smallest size $d$ (often phrased as $\mathbf{X}$ being *in general position*) implies that volume sampling wrt any size $s \geq d$ has full support.

Surprisingly by combining theorems 3 and 4, we can obtain a "covariance type formula" for the pseudo-inverse matrix estimator:

$$\begin{aligned}
&\mathbb{E}[((\mathbf{X}\mathbf{I}_S)^+ - \mathbb{E}[(\mathbf{X}\mathbf{I}_S)^+])^\top\,((\mathbf{X}\mathbf{I}_S)^+ - \mathbb{E}[(\mathbf{X}\mathbf{I}_S)^+])] \\
&= \mathbb{E}[(\mathbf{X}\mathbf{I}_S)^{+\top}(\mathbf{X}\mathbf{I}_S)^+] - \mathbb{E}[(\mathbf{X}\mathbf{I}_S)^+]^\top\,\mathbb{E}[(\mathbf{X}\mathbf{I}_S)^+] \\
&= \frac{n-d+1}{s-d+1}\,\mathbf{X}^{+\top}\mathbf{X}^+ - \mathbf{X}^{+\top}\mathbf{X}^+ = \frac{n-s}{s-d+1}\,\mathbf{X}^{+\top}\mathbf{X}^+. \qquad (1)
\end{aligned}$$

Theorem 4 can also be used to obtain an expectation formula for the Frobenius norm $\|(\mathbf{X}\mathbf{I}_S)^+\|_F$ of the estimator:

$$\mathbb{E}\|(\mathbf{X}\mathbf{I}_S)^+\|_F^2 = \mathbb{E}[\mathrm{tr}((\mathbf{X}\mathbf{I}_S)^{+\top}(\mathbf{X}\mathbf{I}_S)^+)] = \frac{n-d+1}{s-d+1}\|\mathbf{X}^+\|_F^2. \qquad (2)$$

This norm formula has been shown in [1], with numerous applications. Theorem 4 can be viewed as a much stronger pre trace version of the norm formula. Also our proof techniques are quite different

and much simpler. Note that if size $s$ volume sampling for $\mathbf{X}$ does not have full support then (1) becomes a semi-definite inequality $\preceq$ between matrices and (2) an inequality between numbers.

**Proof of Theorem 3** We apply Lemma 1 with $\mathbf{F}(S) = (\mathbf{X}\mathbf{I}_S)^+$. It suffices to show $\mathbf{F}(S) = \sum_{i \in S} P(S_{-i}|S)\mathbf{F}(S_{-i})$ for $P(S_{-i}|S) := \frac{1 - \mathbf{x}_i^\top (\mathbf{X}_S \mathbf{X}_S^\top)^{-1} \mathbf{x}_i}{s - d}$, ie:

$$(\mathbf{X}\mathbf{I}_S)^+ = \sum_{i \in S} \frac{1 - \mathbf{x}_i^\top (\mathbf{X}_S \mathbf{X}_S^\top)^{-1} \mathbf{x}_i}{s - d} \underbrace{(\mathbf{X}\mathbf{I}_{S_{-i}})^+}_{(\mathbf{X}\mathbf{I}_{S_{-i}})^\top (\mathbf{X}_{S_{-i}} \mathbf{X}_{S_{-i}}^\top)^{-1}}.$$

Proven by applying Sherman Morrison to $(\mathbf{X}_{S_{-i}} \mathbf{X}_{S_{-i}}^\top)^{-1} = (\mathbf{X}_S \mathbf{X}_S^\top - \mathbf{x}_i \mathbf{x}_i^\top)^{-1}$ on the rhs:

$$\sum_i \frac{1 - \mathbf{x}_i^\top (\mathbf{X}_S \mathbf{X}_S^\top)^{-1} \mathbf{x}_i}{s - d} \; ((\mathbf{X}\mathbf{I}_S)^\top - \mathbf{e}_i \mathbf{x}_i^\top) \left( (\mathbf{X}_S \mathbf{X}_S^\top)^{-1} + \frac{(\mathbf{X}_S \mathbf{X}_S^\top)^{-1} \mathbf{x}_i \mathbf{x}_i^\top (\mathbf{X}_S \mathbf{X}_S^\top)^{-1}}{1 - \mathbf{x}_i^\top (\mathbf{X}_S \mathbf{X}_S^\top)^{-1} \mathbf{x}_i} \right).$$

We now expand the last two factors into 4 terms. The expectation of the first $(\mathbf{X}\mathbf{I}_S)^\top (\mathbf{X}_S \mathbf{X}_S^\top)^{-1}$ is $(\mathbf{X}\mathbf{I}_S)^+$ (which is the lhs) and the expectations of the remaining three terms times $s - d$ sum to 0:

$$-\sum_{i \in S} (1 - \mathbf{x}_i^\top (\mathbf{X}_S \mathbf{X}_S^\top)^{-1} \mathbf{x}_i) \, \mathbf{e}_i \mathbf{x}_i^\top (\mathbf{X}_S \mathbf{X}_S^\top)^{-1} + (\mathbf{X}\mathbf{I}_S)^\top (\mathbf{X}_S \mathbf{X}_S^\top)^{-1} \sum_{i \in S} \mathbf{x}_i \mathbf{x}_i^\top (\mathbf{X}_S \mathbf{X}_S^\top)^{-1}$$

$$-\sum_{i \in S} \mathbf{e}_i (\mathbf{x}_i^\top (\mathbf{X}_S \mathbf{X}_S^\top)^{-1} \mathbf{x}_i) \, \mathbf{x}_i^\top (\mathbf{X}_S \mathbf{X}_S^\top)^{-1} = 0. \qquad \blacksquare$$

**Proof of Theorem 4** Choose $\mathbf{F}(S) = \frac{s - d + 1}{n - d + 1} (\mathbf{X}_S \mathbf{X}_S^\top)^{-1}$. By Lemma 1 it suffices to show $\mathbf{F}(S) = \sum_{i \in S} P(S_{-i}|S)\mathbf{F}(S_{-i})$ for volume sampling:

$$\frac{s - d + 1}{n - d + 1} (\mathbf{X}_S \mathbf{X}_S^\top)^{-1} = \sum_{i \in S} \frac{1 - \mathbf{x}_i^\top (\mathbf{X}_S \mathbf{X}_S^\top)^{-1} \mathbf{x}_i}{s - d} \frac{s - d}{n - d + 1} (\mathbf{X}_{S_{-i}} \mathbf{X}_{S_{-i}}^\top)^{-1}$$

To show this we apply Sherman Morrison to $(\mathbf{X}_{S_{-i}} \mathbf{X}_{S_{-i}}^\top)^{-1}$ on the rhs:

$$\sum_{i \in S} (1 - \mathbf{x}_i^\top (\mathbf{X}_S \mathbf{X}_S^\top)^{-1} \mathbf{x}_i) \left( (\mathbf{X}_S \mathbf{X}_S^\top)^{-1} + \frac{(\mathbf{X}_S \mathbf{X}_S^\top)^{-1} \mathbf{x}_i \mathbf{x}_i^\top (\mathbf{X}_S \mathbf{X}_S^\top)^{-1}}{1 - \mathbf{x}_i^\top (\mathbf{X}_S \mathbf{X}_S^\top)^{-1} \mathbf{x}_i} \right)$$

$$= (s - d)(\mathbf{X}_S \mathbf{X}_S^\top)^{-1} + (\mathbf{X}_S \mathbf{X}_S^\top)^{-1} \sum_{i \in S} \mathbf{x}_i \mathbf{x}_i^\top (\mathbf{X}_S \mathbf{X}_S^\top)^{-1} = (s - d + 1)\,(\mathbf{X}_S \mathbf{X}_S^\top)^{-1}.$$

If some denominators $1 - \mathbf{x}_i^\top (\mathbf{X}_S \mathbf{X}_S^\top)^{-1} \mathbf{x}_i$ are zero, then only sum over $i$ for which the denominators are positive. In this case the above matrix equality becomes a positive-definite inequality $\preceq$. $\qquad \blacksquare$

# 4  Linear regression with few labels

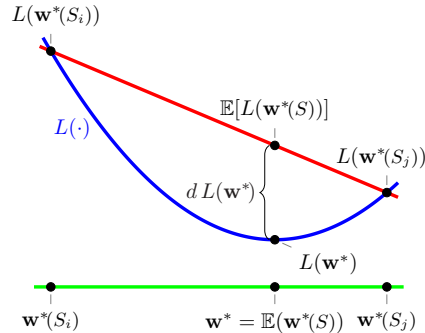

Our main motivation for studying volume sampling came from asking the following simple question. Suppose we want to solve a $d$-dimensional linear regression problem with a matrix $\mathbf{X} \in \mathbb{R}^{d \times n}$ of input column vectors and a label vector $\mathbf{y} \in \mathbb{R}^n$, ie find $\mathbf{w} \in \mathbb{R}^d$ that minimizes the least squares loss $L(\mathbf{w}) = \|\mathbf{X}^\top \mathbf{w} - \mathbf{y}\|^2$:

$$\mathbf{w}^* \overset{\text{def}}{=} \underset{\mathbf{w} \in \mathbb{R}^d}{\arg\min}\, L(\mathbf{w}) = \mathbf{X}^{+\top} \mathbf{y},$$

**Figure 2:** Unbiased estimator $\mathbf{w}^*(S)$ in expectation suffers loss $(d + 1)\, L(\mathbf{w}^*)$.

but the access to label vector $\mathbf{y}$ is restricted. We are allowed to pick a subset $S \subseteq \{1..n\}$ for which the labels $y_i$ (where $i \in S$) are revealed to us, and then solve the subproblem $(\mathbf{X}_S, \mathbf{y}_S)$, obtaining $\mathbf{w}^*(S)$. What is the smallest number of labels such that for any $\mathbf{X}$, we can find $\mathbf{w}^*(S)$ for which $L(\mathbf{w}^*(S))$ is only a multiplicative factor away from $L(\mathbf{w}^*)$ (independent of the number of input vectors $n$)? This question was posed as an open problem by [3]. It is easy to show that we need at least $d$ labels (when $\mathbf{X}$ is full-rank), so as to guarantee the uniqueness of solution $\mathbf{w}^*(S)$. We use volume sampling to show that $d$ labels are in fact sufficient (proof in Section 4.1).

**Theorem 5** *If the input matrix $\mathbf{X} \in \mathbb{R}^{d \times n}$ is in general position, then for any label vector $\mathbf{y} \in \mathbb{R}^n$, the expected square loss (on all $n$ labeled vectors) of the optimal solution $\mathbf{w}^*(S)$ for the subproblem $(\mathbf{X}_S, \mathbf{y}_S)$, with the $d$-element set $S$ obtained from volume sampling, is given by*

$$\mathbb{E}[L(\mathbf{w}^*(S))] = (d+1)\ L(\mathbf{w}^*).$$

*If $\mathbf{X}$ is not in general position, then the expected loss is upper-bounded by $(d+1)\ L(\mathbf{w}^*)$.*

The factor $d+1$ cannot be improved when selecting only $d$ labels (we omit the proof):

**Proposition 6** *For any $d$, there exists a least squares problem $(\mathbf{X}, \mathbf{y})$ with $d+1$ vectors in $\mathbb{R}^d$ such that for every $d$-element index set $S \subseteq \{1, ..., d+1\}$, we have*

$$L(\mathbf{w}^*(S)) = (d+1)\ L(\mathbf{w}^*).$$

Note that the multiplicative factor in Theorem 5 does not depend on $n$. It is easy to see that this cannot be achieved by any deterministic algorithm (without the access to labels). Namely, suppose that $d = 1$ and $\mathbf{X}$ is a vector of all ones, whereas the label vector $\mathbf{y}$ is a vector of all ones except for a single zero. No matter which column index we choose deterministically, if that index corresponds to the label 0, the solution to the subproblem will incur loss $L(\mathbf{w}^*(S)) = n\ L(\mathbf{w}^*)$. The fact that volume sampling is a joint distribution also plays an essential role in proving Theorem 5. Consider a matrix $\mathbf{X}$ with exactly $d$ unique linearly independent columns (and an arbitrary number of duplicates). Any iid column sampling distribution (like for example leverage score sampling) will require $\Omega(d \log d)$ samples to retrieve all $d$ unique columns (ie coupon collector problem), which is necessary to get any multiplicative loss bound.

The exact expectation formula for the least squares loss under volume sampling suggests a deep connection between linear regression and this distribution. We can use Theorem 3 to further strengthen that connection. Note, that the least squares estimator obtained through volume sampling can be written as $\mathbf{w}^*(S) = (\mathbf{X}\mathbf{I}_S)^{+\top}\mathbf{y}$. Applying formula for the expectation of pseudo-inverse, we conclude that $\mathbf{w}^*(S)$ is an unbiased estimator of $\mathbf{w}^*$.

**Proposition 7** *Let $\mathbf{X} \in \mathbb{R}^{d \times n}$ be a full-rank matrix and $n \geq s \geq d$. Let $S \subseteq 1..n$ be a size $s$ volume sampled set over $\mathbf{X}$. Then, for arbitrary label vector $\mathbf{y} \in \mathbb{R}^n$, we have*

$$\mathbb{E}[\mathbf{w}^*(S)] = \mathbb{E}[(\mathbf{X}\mathbf{I}_S)^{+\top}\mathbf{y}] = \mathbf{X}^{+\top}\mathbf{y} = \mathbf{w}^*.$$

For size $s = d$ volume sampling, the fact that $\mathbb{E}[\mathbf{w}^*(S)]$ equals $\mathbf{w}^*$ can be found in an early paper [2]. They give a direct proof based on Cramer's rule. For us the above proposition is a direct consequence of the matrix expectation formula given in Theorem 3 that holds for volume sampling of any size $s \geq d$. In contrast, the loss expectation formula of Theorem 5 is limited to sampling of size $s = d$. Bounding the loss expectation for $s > d$ remains an open problem. However, we consider a different strategy for extending volume sampling in linear regression. Combining Proposition 7 with Theorem 5 we can compute the variance of predictions generated by volume sampling, and obtain tighter multiplicative loss bounds by sampling multiple $d$-element subsets $S_1, ..., S_t$ independently.

**Theorem 8** *Let $(\mathbf{X}, \mathbf{y})$ be as in Theorem 5. For $k$ independent size $d$ volume samples $S_1, ..., S_k$,*

$$\mathbb{E}\left[L\left(\frac{1}{k}\sum_{j=1}^{k}\mathbf{w}^*(S_j)\right)\right] = \left(1 + \frac{d}{k}\right) L(\mathbf{w}^*).$$

**Proof** Denote $\widehat{\mathbf{y}} \stackrel{\text{def}}{=} \mathbf{X}^\top\mathbf{w}^*$ and $\widehat{\mathbf{y}}(S) \stackrel{\text{def}}{=} \mathbf{X}^\top\mathbf{w}^*(S)$ as the predictions generated by $\mathbf{w}^*$ and $\mathbf{w}^*(S)$ respectively. We perform bias-variance decomposition of the loss of $\mathbf{w}^*(S)$ (for size $d$ volume sampling):

$$\begin{aligned}
\mathbb{E}[L(\mathbf{w}^*(S))] &= \mathbb{E}[\|\widehat{\mathbf{y}}(S) - \mathbf{y}\|^2] = \mathbb{E}[\|\widehat{\mathbf{y}}(S) - \widehat{\mathbf{y}} + \widehat{\mathbf{y}} - \mathbf{y}\|^2] \\
&= \mathbb{E}[\|\widehat{\mathbf{y}}(S) - \widehat{\mathbf{y}}\|^2] + \mathbb{E}[2(\widehat{\mathbf{y}}(S) - \widehat{\mathbf{y}})^\top(\widehat{\mathbf{y}} - \mathbf{y})] + \|\widehat{\mathbf{y}} - \mathbf{y}\|^2 \\
&\stackrel{(*)}{=} \sum_{i=1}^{n}\mathbb{E}\left[(\widehat{y}(S)_i - \mathbb{E}[\widehat{y}(S)_i])^2\right] + L(\mathbf{w}^*) = \sum_{i=1}^{n}\text{Var}[\widehat{y}(S)_i] + L(\mathbf{w}^*),
\end{aligned}$$

where $(*)$ follows from Theorem 3. Now, we use Theorem 5 to obtain the total variance of predictions:

$$\sum_{i=1}^{n} \text{Var}[\widehat{y}(S)_i] = \mathbb{E}[L(\mathbf{w}^*(S))] - L(\mathbf{w}^*) = d\, L(\mathbf{w}^*).$$

Now the expected loss of the average weight vector wrt sampling $k$ independent sets $S_1, ..., S_k$ is:

$$\mathbb{E}\left[ L\left( \frac{1}{k} \sum_{j=1}^{k} \mathbf{w}^*(S_j) \right) \right] = \sum_{i=1}^{n} \text{Var}\left[ \frac{1}{k} \sum_{j=1}^{k} \widehat{y}(S_j)_i \right] + L(\mathbf{w}^*)$$

$$= \frac{1}{k^2} \left( \sum_{j=1}^{k} d\, L(\mathbf{w}^*) \right) + L(\mathbf{w}^*) = \left( 1 + \frac{d}{k} \right) L(\mathbf{w}^*). \qquad \blacksquare$$

It is worth noting that the average weight vector used in Theorem 8 is not expected to perform better than taking the solution to the joint subproblem, $\mathbf{w}^*(S_{1:k})$, where $S_{1:k} = S_1 \cup ... \cup S_k$. However, theoretical guarantees for that case are not yet available.

## 4.1 Proof of Theorem 5

We use the following lemma regarding the leave-one-out loss for linear regression [4]:

**Lemma 9** *Let* $\mathbf{w}^*(-i)$ *denote the least squares solution for problem* $(\mathbf{X}_{-i}, \mathbf{y}_{-i})$. *Then, we have*

$$L(\mathbf{w}^*) = L(\mathbf{w}^*(-i)) - \mathbf{x}_i^\top (\mathbf{X}\mathbf{X}^\top)^{-1} \mathbf{x}_i\, \ell_i(\mathbf{w}^*(-i)), \quad \text{where} \quad \ell_i(\mathbf{w}) \overset{\text{def}}{=} (\mathbf{x}_i^\top \mathbf{w} - y_i)^2.$$

When $\mathbf{X}$ has $d+1$ columns and $\mathbf{X}_{-i}$ is a full-rank $d \times d$ matrix, then $L(\mathbf{w}^*(-i)) = \ell_i(\mathbf{w}^*(-i))$ and Lemma 9 leads to the following:

$$\det(\widetilde{\mathbf{X}}\widetilde{\mathbf{X}}^\top) \overset{(1)}{=} \det(\mathbf{X}\mathbf{X}^\top) \overbrace{\|\widehat{\mathbf{y}} - \mathbf{y}\|^2}^{L(\mathbf{w}^*)} \qquad \text{where } \widetilde{\mathbf{X}} = \begin{pmatrix} \mathbf{X} \\ \mathbf{y}^\top \end{pmatrix}$$

$$\overset{(2)}{=} \det(\mathbf{X}\mathbf{X}^\top)(1 - \mathbf{x}_i^\top (\mathbf{X}\mathbf{X}^\top)^{-1} \mathbf{x}_i) \ell_i(\mathbf{w}^*(-i))$$

$$\overset{(3)}{=} \det(\mathbf{X}_{-i}\mathbf{X}_{-i}^\top) \ell_i(\mathbf{w}^*(-i)), \qquad (3)$$

where (1) is the "base $\times$ height" formula for volume, (2) follows from Lemma 9 and (3) follows from a standard determinant formula. Returning to the proof, our goal is to find the expected loss $\mathbb{E}[L(\mathbf{w}^*(S))]$, where $S$ is a size $d$ volume sampled set. First, we rewrite the expectation as follows:

$$\mathbb{E}[L(\mathbf{w}^*(S))] = \sum_{S, |S|=d} P(S) L(\mathbf{w}^*(S)) = \sum_{S, |S|=d} P(S) \sum_{j=1}^{n} \ell_j(\mathbf{w}^*(S))$$

$$= \sum_{S, |S|=d} \sum_{j \notin S} P(S)\, \ell_j(\mathbf{w}^*(S)) = \sum_{T, |T|=d+1} \sum_{j \in T} P(T_{-j})\, \ell_j(\mathbf{w}^*(T_{-j})). \qquad (4)$$

We now use (3) on the matrix $\mathbf{X}_T$ and test instance $\mathbf{x}_j$ (assuming $\text{rank}(\mathbf{X}_{T_{-j}}) = d$):

$$P(T_{-j})\, \ell_j(\mathbf{w}^*(T_{-j})) = \frac{\det(\mathbf{X}_{T_{-j}}\mathbf{X}_{T_{-j}}^\top)}{\det(\mathbf{X}\mathbf{X}^\top)}\, \ell_j(\mathbf{w}^*(T_{-j})) = \frac{\det(\widetilde{\mathbf{X}}_T \widetilde{\mathbf{X}}_T^\top)}{\det(\mathbf{X}\mathbf{X}^\top)}. \qquad (5)$$

Since the summand does not depend on the index $j \in T$, the inner summation in (4) becomes a multiplication by $d+1$. This lets us write the expected loss as:

$$\mathbb{E}[L(\mathbf{w}^*(S))] = \frac{d+1}{\det(\mathbf{X}\mathbf{X}^\top)} \sum_{T, |T|=d+1} \det(\widetilde{\mathbf{X}}_T \widetilde{\mathbf{X}}_T^\top) \overset{(1)}{=} (d+1) \frac{\det(\widetilde{\mathbf{X}}\widetilde{\mathbf{X}}^\top)}{\det(\mathbf{X}\mathbf{X}^\top)} \overset{(2)}{=} (d+1)\, L(\mathbf{w}^*),$$

$$(6)$$

where (1) follows from the Cauchy-Binet formula and (2) is an application of the "base $\times$ height" formula. If $\mathbf{X}$ is not in general position, then for some summands in (5), $\text{rank}(\mathbf{X}_{T_{-j}}) < d$ and $P(T_{-j}) = 0$. Thus the left-hand side of (5) is 0, while the right-hand side is non-negative, so (6) becomes an inequality, completing the proof of Theorem 5.

# 5 Efficient algorithm for volume sampling

In this section we propose an algorithm for efficiently performing exact volume sampling for any $s \geq d$. This addresses the question posed by [1], asking for a polynomial-time algorithm for the case when $s > d$. [6, 11] gave an algorithm for the case when $s = d$, which runs in time $O(nd^3)$. Recently, [16] offered an algorithm for arbitrary $s$, which has complexity $O(n^4 s)$. We propose a new method, which uses our techniques to achieve the time complexity $O((n - s + d)nd)$, a direct improvement over [16] by a factor of at least $n^2$. Our algorithm also offers an improvement for $s = d$ in certain regimes. Namely, when $n = o(d^2)$, then our algorithm runs in time $O(n^2 d) = o(nd^3)$, faster than the method proposed by [6].

Our algorithm implements reverse iterative sampling from Corollary 2. After removing $q$ columns, we are left with an index set of size $n - q$ that is distributed according to volume sampling for column set size $n - q$.

**Theorem 10** *The sampling algorithm runs in time $O((n - s + d)nd)$, using $O(d^2 + n)$ additional memory, and returns set $S$ which is distributed according to size $s$ volume sampling over $\mathbf{X}$.*

**Proof** For correctness we show the following invariants that hold at the beginning of the **while** loop:

$$p_i = 1 - \mathbf{x}_i^\top (\mathbf{X}_S \mathbf{X}_S^\top)^{-1} \mathbf{x}_i = (|S| - d) \, P(S_{-i}|S) \qquad \text{and} \qquad \mathbf{Z} = (\mathbf{X}_S \mathbf{X}_S^\top)^{-1}.$$

At the first iteration the invariants trivially hold. When updating the $p_j$ we use $\mathbf{Z}$ and the $p_i$ from the previous iteration, so we can rewrite the update as

---

**Reverse iterative volume sampling**

---

**Input:** $\mathbf{X} \in \mathbb{R}^{d \times n}, \, s \in \{d..n\}$
$\mathbf{Z} \leftarrow (\mathbf{X}\mathbf{X}^\top)^{-1}$
$\forall_{i \in \{1..n\}} \quad p_i \leftarrow 1 - \mathbf{x}_i^\top \mathbf{Z} \mathbf{x}_i$
$S \leftarrow \{1, .., n\}$
**while** $|S| > s$
    Sample $i \propto p_i$ out of $S$
    $S \leftarrow S - \{i\}$
    $\mathbf{v} \leftarrow \mathbf{Z}\mathbf{x}_i / \sqrt{p_i}$
    $\forall_{j \in S} \quad p_j \leftarrow p_j - (\mathbf{x}_j^\top \mathbf{v})^2$
    $\mathbf{Z} \leftarrow \mathbf{Z} + \mathbf{v}\mathbf{v}^\top$
**end**
**return** $S$

---

$$
\begin{aligned}
p_j &\leftarrow p_j - (\mathbf{x}_j^\top \mathbf{v})^2 \\
&= 1 - \mathbf{x}_j^\top (\mathbf{X}_S \mathbf{X}_S^\top)^{-1} \mathbf{x}_j - \frac{(\mathbf{x}_j^\top \mathbf{Z} \mathbf{x}_i)^2}{1 - \mathbf{x}_i^\top (\mathbf{X}_S \mathbf{X}_S^\top)^{-1} \mathbf{x}_i} \\
&= 1 - \mathbf{x}_j^\top (\mathbf{X}_S \mathbf{X}_S^\top)^{-1} \mathbf{x}_j - \frac{\mathbf{x}_j^\top (\mathbf{X}_S \mathbf{X}_S^\top)^{-1} \mathbf{x}_i \mathbf{x}_i^\top (\mathbf{X}_S \mathbf{X}_S^\top)^{-1} \mathbf{x}_j}{1 - \mathbf{x}_i^\top (\mathbf{X}_S \mathbf{X}_S^\top)^{-1} \mathbf{x}_i} \\
&= 1 - \mathbf{x}_j^\top \left( (\mathbf{X}_S \mathbf{X}_S^\top)^{-1} + \frac{(\mathbf{X}_S \mathbf{X}_S^\top)^{-1} \mathbf{x}_i \mathbf{x}_i^\top (\mathbf{X}_S \mathbf{X}_S^\top)^{-1}}{1 - \mathbf{x}_i^\top (\mathbf{X}_S \mathbf{X}_S^\top)^{-1} \mathbf{x}_i} \right) \mathbf{x}_j \\
&\overset{(*)}{=} 1 - \mathbf{x}_j^\top (\mathbf{X}_{S_{-i}} \mathbf{X}_{S_{-i}}^\top)^{-1} \mathbf{x}_j = (|S| - 1 - d) \, P(S_{-i,j}|S_{-i}),
\end{aligned}
$$

where $(*)$ follows from the Sherman-Morrison formula. The update of $\mathbf{Z}$ is also an application of Sherman-Morrison and this concludes the proof of correctness.

Runtime: Computing the initial $\mathbf{Z} = (\mathbf{X}\mathbf{X}^\top)^{-1}$ takes $O(nd^2)$, as does computing the initial values of $p_j$'s. Inside the **while** loop, updating $p_j$'s takes $O(|S|d) = O(nd)$ and updating $\mathbf{Z}$ takes $O(d^2)$. The overall runtime becomes $O(nd^2 + (n - s)nd) = O((n - s + d)nd)$. The space usage (in addition to the input data) is dominated by the $p_i$ values and matrix $\mathbf{Z}$. ∎

# 6 Conclusions

We developed exact formulas for $\mathbb{E}[(\mathbf{X}\mathbf{I}_S)^+)]$ and $\mathbb{E}[(\mathbf{X}\mathbf{I}_S)^+)^2]$ when the subset $S$ of $s$ column indices is sampled proportionally to the volume $\det(\mathbf{X}_S \mathbf{X}_S^\top)$. The formulas hold for any fixed size $s \in \{d..n\}$. These new expectation formulas imply that the solution $\mathbf{w}^*(S)$ for a volume sampled subproblem of a linear regression problem is unbiased. We also gave a formula relating the loss of the subproblem to the optimal loss (ie $\mathbb{E}(L(\mathbf{w}^*(S))) = (d + 1)L(\mathbf{w}^*)$). However, this result only holds for sample size $s = d$. It is an open problem to obtain such an exact expectation formula for $s > d$.

A natural algorithm is to draw $k$ samples $S_i$ of size $d$ and return $\mathbf{w}^*(S_{1:k})$, where $S_{1:k} = \bigcup_i S_i$. We were able to get exact expressions for the loss $L(\frac{1}{k} \sum_i \mathbf{w}^*(S_i))$ of the average predictor but it is an open problem to get nontrivial bounds for the loss of the best predictor $\mathbf{w}^*(S_{1:k})$.

We were able to show that for small sample sizes, volume sampling a set jointly has the advantage: It achieves a multiplicative bound for the smallest sample size $d$, whereas any independent sampling routine requires sample size at least $\Omega(d \log d)$.

We believe that our results demonstrate a fundamental connection between volume sampling and linear regression, which demands further exploration. Our loss expectation formula has already been applied by [12] to the task of linear regression without correspondence.

**Acknowledgements** Thanks to Daniel Hsu and Wojciech Kotłowski for many valuable discussions. This research was supported by NSF grant IIS-1619271.

## Footnotes

[1]Note that $\frac{0}{0}$ determinant ratios are avoided along the path because paths with such ratios always lead to sets of probability 0 and in the corollary we only consider paths to nodes $S$ for which $\det(\mathbf{X}_S\mathbf{X}_S) > 0$.

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
