[Reviews · NeurIPS 2017]

Reviewer 1



The authors make connection between volume sampling and linear regression (matrix pseudo-inverses). The main contribution is that volume sampling is an unbiased estimator to the computation of matrix pseudo-inverses. Moreover, an application of their main contribution is applied to the case of linear regression with few samples. The paper is well-written and the main contributions are very well presented. The only concern that I have is the applicability of the results. That said, I find the main contribution of the paper very interesting. Major comments: 1) Is it possible to give concentration bounds on the number of samples required for some approximation to X^{+}? Minor comments: *) Theorem 5, L161: Please define "general position", the term is (probably) only clear to a computational geometry audience.

Reviewer 2



I could go either way on this paper, though am slightly positive. The short summary is that the submission gives elegant expectation bounds with non-trivial arguments, but if one wants constant factor approximations (or 1+eps)-approximations), then existing algorithms are faster and read fewer labels. So it's unclear to me if there is a solid application of the results in the paper. In more detail: On the positive side it's great to see an unbiased estimator of the pseudoinverse by volume sampling, which by linearity gives an unbiased estimator to the least squares solution vector. I haven't seen such a statement before. It's also nice to see an unbiased estimator of the least squares loss function when exactly d samples are taken. The authors use an inductive argument, with determinants canceling in the induction, and Sherman-Morrison. They give a simple proof of Cauchy-Binet as well. The n^2 speedup the authors give for exact volume sampling over previous work is also important. On the negative side, since their exact volume sampling is nd^2 time for a single volume sample (as opposed to approximate volume sampling which is faster), it seems there are more efficient solutions for these problems, in particular for standard least squares. The authors don't claim improvement for least squares, but rather they consider the case when reading the labels to the examples is expensive and they only want to read a small number s of labels. The authors are right in that leverage score sampling would require reading d log d labels to get any relative error approximation, and although one could use Batson-Spielman-Srivastava on top of this, I think this would still read d log d labels instead of the d labels that volume sampling uses (even though Batson-Spielman-Srivastava would end up with d sampled columns, to choose these d columns as far as I know would need to read at least d log d labels. This would be worth discussing / fleshing out). But, the authors only get a d-approximation by doing this and need to take d independent volume samples, or d^2 total samples, to get a constant factor approximation to regression. On the other hand, leverage score sampling with d log d samples would get a constant factor approximation. So regarding applications, if one asks the question - with any amount of time complexity, how can I read as few labels as possible to get an O(d)-approximation, then I think the authors give the first algorithm for this problem. However if one wants a constant approximation or 1+eps-approximation, previous work seems to give faster algorithms and fewer samples. It would be good if the authors compare their guarantees / running times to others for estimating the pseudoinverse and optimal regression vector, such as the recent ones in: Price, Song, Woodruff, Fast Regression with an $\ell_\infty$ Guarantee, ICALP, 2017 and the earlier ones for estimating the regression solution vector in: Sarlos, Improved Approximation Algorithms for Large Matrices via Random Projections, FOCS, 2006 (the running time should be combined with more modern input sparsity time subspace embeddings) I have read the rebuttal and am still positive on the paper.

Reviewer 3



Summary: This paper studies the following problem: Given an n X d matrix A with n >> d, approximate the pseudoinverse of A by sampling columns and computing the pseudoinverse of the subsampled matrix. The main result is that by doing volume sampling: sample a set S with probability proportional to A_S A^T_S, we get an unbiased estimator for the pseudoinverse. In particular, the authors apply this to linear regression to show that if L(w*) is the optimal loss and L(w_S) is the loss obtained by using a sample of size s=d, then E[L(w_S)] = (d+1)L(w*). This is a difference from i.i.d. sampling where one needs to sample d log d columns before a multiplicative error bound can be achieved. The authors show that by computing the variance of the estimator, one can take many random samples and get tight bounds on the loss of the average solution. Using their ideas, the authors also give an improved algorithm to do volume sampling for s > d. Their new algorithm is a quadratic improvement over previous results. This is a very well written paper with a significant advance over prior results in the area. The one unsatisfying aspect of the result is that currently the authors can't prove much better bounds on the expected loss when sampling s > d columns. The authors should do a better job of mentioning what exactly breaks down in their analysis and how significant a challenge is it to achieve such bounds. Overall, the paper is a good fit for publication to NIPS and will be of interest to many participants.